# Dissipative quadratic soliton in the cascaded nonlinearity limit

Mingming Nie [1,2] ✉, Jonathan Musgrave [1] & Shu-Wei Huang [1,3] ✉

Dissipative quadratic solitons (DQSs), enabled by nonlinearity engineering through cascaded quadratic processes, have remained a central theoretical prediction in nonlinear optics since their proposal in 1997. Despite their predicted ultralow operational thresholds, remarkable tunability, and potential for spectral extension into unconventional wavelengths, their experimental realization has remained elusive. Here, we demonstrate bright, dual-color DQS generation in the normal dispersion regime, in strong agreement with the proposed theoretical framework. Furthermore, by simply adjusting the nonlinear crystal temperature - without any structural modification - we reverse the sign of the effective nonlinearity and switch from bright DQS to platicon generation in situ. This work not only advances the fundamental understanding of dissipative solitons but also establishes a practical pathway for ultralow-threshold frequency comb generation at unconventional wavelengths, with broad implications for applications such as atomic clocks, optical coherence tomography, and astrocombs.

A dissipative Kerr soliton (DKS) is a stable, localized wave packet that forms in a dispersion-engineered nonlinear resonator through a double balance between dispersion and Kerr nonlinearity, as well as parametric gain and cavity loss[1,2]. DKS has drawn significant attention for its ability to generate coherent optical frequency combs, enabling transformative advancements across a wide range of fields. Notable applications with DKS microcombs include highly multiplexed coherent optical communications[3,4], astrocombs for precise astronomical spectrograph calibration[5,6], coherent ranging for high-accuracy distance measurement[7,8], dual-comb spectroscopy for fast and precise chemical analysis[9,10], integrated frequency synthesizers[11,12], and optical clock systems[13–15]. Furthermore, their compact size, low power consumption, and potential for on-chip integration make them ideal for portable and space-based technologies.

It has been demonstrated that cascaded quadratic processes can create an effective Kerr nonlinearity (EKN)[16], whose feature can be flexibly controlled via phase mismatch. When both phase mismatch and group velocity mismatch (GVM) are close to zero, the EKN dominates over the material Kerr nonlinearity (MKN). In this regime, EKN becomes the primary nonlinearity governing dissipative soliton formation

dynamics, giving rise to the concept of dissipative quadratic solitons (DQSs)[17–24]. Notably, the maximum EKN occurs near the zero phase mismatch point, and its sign can be easily reversed by tuning across this point. This enables a wide range of EKN tuning, from negative to positive, allowing it to balance any dispersion for soliton generation.

In a different regime characterized by large GVM, walk-off-induced DQS has been observed in a synchronously pumped optical parametric oscillator[25]. This form of DQS arises from the interplay among dispersion, gain saturation, timing mismatch, and gain clamping, resulting in a sensitivity to the pump pulse duration. This behavior contrasts with the canonical DQS in the cascaded nonlinearity limit, where the EKN predominantly governs soliton formation dynamics.

Compared to dispersion-engineered DKS, nonlinearity-engineered DQS offers greater flexibility in selecting the comb spectral range and provides in situ control over the DQS characteristics. Additionally, the large EKN−typically 100 times greater than the MKN−significantly reduces the threshold power for DQS generation, making it much lower than that required for DKS formation. Finally, since EKN originates from the two-wave interaction in the quadratically nonlinear crystal, DQS is intrinsically a pair of dual-color mode-locked pulses and can be easily

[1]Department of Electrical, Computer and Energy Engineering, University of Colorado Boulder, Boulder, Colorado, USA. [2]Key Laboratory of Optical Fiber Sensing and Communications (Education Ministry of China), University of Electronic Science and Technology of China, Chengdu, China. [3]3DIC Research Center, National Yang Ming Chiao Tung University, Hsinchu, Taiwan. ✉e-mail: mingming.nie@uestc.edu.cn; shuwei.huang@colorado.edu

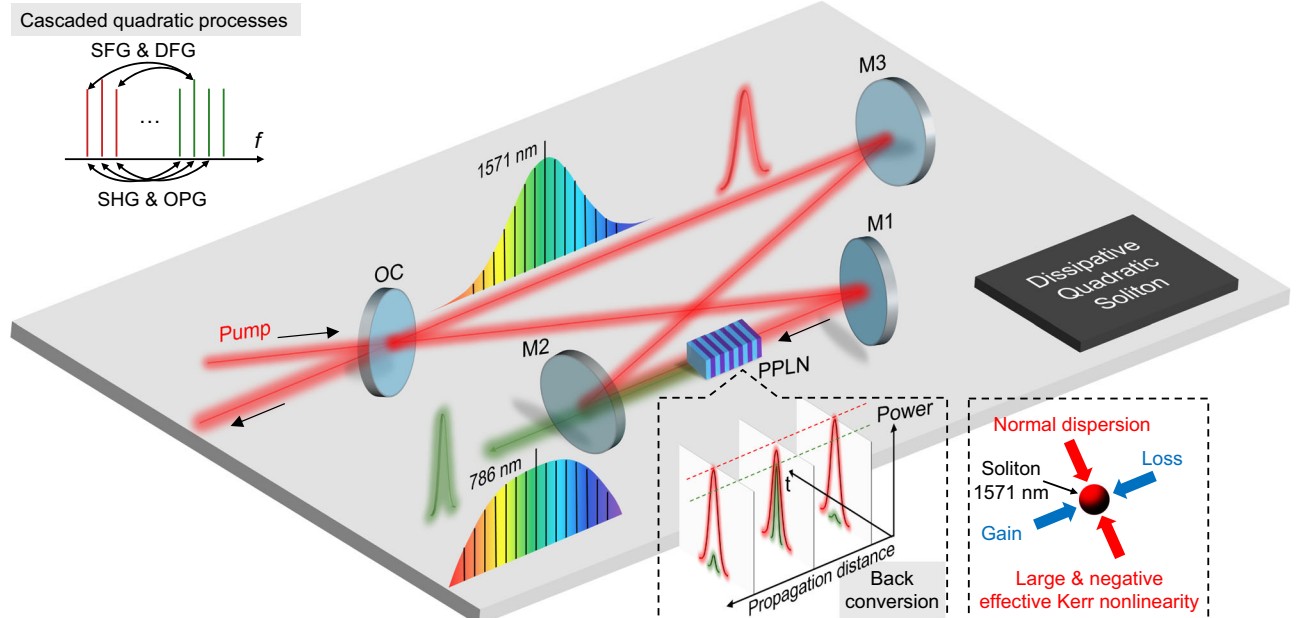

**Fig. 1 | Schematic of DQS comb generation.** The four-mirror free-space cavity incorporating a PPLN crystal is pumped by a 12-ps pulsed laser with *p* polarization. The poling period and the temperature of the PPLN crystal are chosen to satisfy the type-I phase matching condition. Top left corner: diagram of cascaded quadratic processes. Bottom right corner: the left dashed box shows the back conversion between the FF and SH pulses inside the PPLN crystal, leading to synthetic four wave mixing and EKN for DQS comb generation. The right dashed box highlights that the observed bright DQS results from a double balance between normal dispersion and negative EKN, as well as coherent driving and cavity loss. OC: output coupler with transmission of -1.1% at 1571 nm, M1-M3: mirrors with high reflectivity at -1571 nm and high transmission at -786 nm.

extended to form multi-color combs via the efficient intracavity non-linear frequency conversion.

Recent experiments have offered encouraging insights into the development of DQS[26–30]. Using periodically poled lithium niobate (PPLN) crystals, modulation instability (MI) frequency combs have been observed[26–29], and the demonstration of quantum-correlated twin beams has marked a significant milestone[30]. Advancing from MI combs to DQS generation, which promises higher coherence and broader bandwidth, remains a challenging frontier.

In this paper, we demonstrate nonlinearity-engineered DQS in a free-space, singly resonant cavity-enhanced second-harmonic generation (CE-SHG) setup (Fig. 1). A single mean-field equation is derived to describe the DQS formation dynamics, highlighting the origin of the large EKN. The DQS nature is identified through excellently matched simulation and experiment, revealing a dual-color pulse pair with sech²-shaped spectrum for the fundamental field (FF) and a flat-top spectrum for the second harmonic (SH) field. Initially, the phase mismatch is set to a negative value, allowing the highly negative EKN to balance the normal dispersion of the PPLN crystal at 1571 nm, enabling ultralow-threshold bright DQS generation in the normal dispersion regime. By simply adjusting the crystal temperature, we reverse the phase mismatch and consequently the EKN, achieving in situ switching between bright DQSs and platicons, each with distinct spectro-temporal characteristics[31–34]. Additionally, by tuning the pump wavelength and crystal temperature, we demonstrate in situ control over DQS comb properties. Finally, besides SH combs centered at 786 nm, other visible combs centered at 524 nm are also generated through intrinsic sum frequency generation (SFG), offering potential applications in comb self-referencing, optical atomic clocks, and quantum information science[35–39].

## Results
### Operating principles
In the cavity-enhanced second-harmonic generation, quadratic frequency combs arise from cascaded quadratic processes (the upper left corner of Fig. 1), including SHG, optical parametric generation (OPG),

SFG and difference frequency generation (DFG). These back-to-back three-wave mixing processes lead to a synthetic four-wave-mixing process and an effective third-order nonlinearity for the FF. Both the phase mismatch and GVM between the FF and SH field strongly affect the strength and bandwidth of the frequency-dependent effective third-order nonlinearity (see Supplementary Note 1), including the effective Kerr nonlinearity (EKN) and the effective two-photon absorption (ETPA). With near-zero GVM and some phase mismatch, DQS dynamics in the singly resonant cavity-enhanced second-harmonic generation system are well described by the single mean-field equation (SMFE) for the FF $A$ (see Supplementary Note 1):

$$t_R \frac{\partial A}{\partial t} = \left(-\alpha_1 - i\delta_1 - i\frac{k_1'' L}{2}\frac{\partial^2}{\partial \tau^2}\right)A - \alpha_{ETPA}L|A|^2 A + i\gamma_{eff}L|A|^2 A + \sqrt{\theta_1}A_{in}$$

(1)

where $t$ is the "slow time" that describes the envelope evolution over successive roundtrips, $t_R$ is the roundtrip time, $\tau$ is the "fast time" that depicts the temporal profiles in the retarded time frame, and $\alpha_1$ is the total linear cavity loss for the FF, $\delta_1$ is the phase detuning, $k_1''$ is the group velocity dispersion (GVD), $L$ is the nonlinear cavity length, $\theta_1$ is the coupler transmission coefficient, $A_{in}$ is the continuous-wave (CW) pump. $\alpha_{ETPA} = \kappa^2 L sinc^2(\xi/2)/2$ is the ETPA coefficient, $\gamma_{eff} = \kappa^2 L[1 - sinc(\xi)]/\xi$ is the EKN coefficient, where $\kappa$ is the normalized second-order nonlinear coupling coefficient, $\xi = \Delta kL$ is the phase mismatch, and $\Delta k$ is the wave-vector mismatch.

Figure 2a, b show the dependence of both ETPA and EKN with the phase mismatch. When the phase mismatch $\xi = m \cdot 2\pi$ ($m$ is an integer and $m \neq 0$), the ETPA can be neglected namely $\alpha_{ETPA} = 0$ and then Eq. (1) has the exact same form of Lugiato–Lefever equation (LLE) describing the conventional DKS dynamics. Moreover, benefitting from the large quadratic nonlinearity, the EKN can be >100 times larger than the intrinsic MKN at $\xi = \pm 2\pi$ in a 25-mm long type-I phase mismatched PPLN crystal, which will significantly reduce the comb generation threshold. More importantly, the EKN sign can also be engineered by

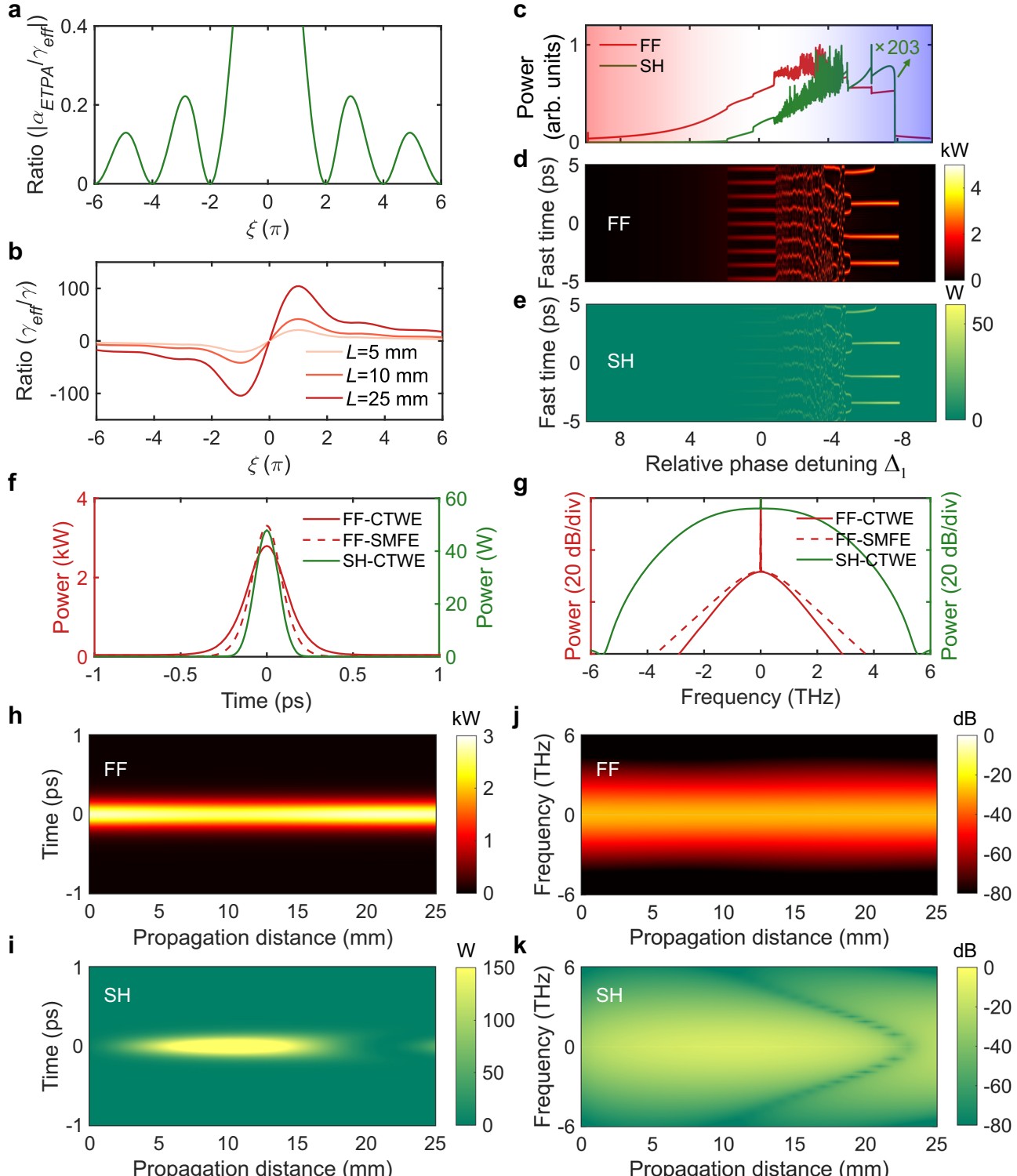

**Fig. 2 | Simulation of bright DQS. a** Effect of phase mismatch on the ratio between ETPA and EKN. **b** Effect of phase mismatch on the ratio between EKN and MKN. **c** Intracavity average power of the FF and SH as a function of normalized detuning $\Delta_1 = \delta_1/\alpha_1$. The SH power is scaled by 203 for clarity. **d** Temporal evolution of the FF as a function of normalized detuning $\Delta_1 = \delta_1/\alpha_1$. **e** Temporal evolution of the SH as a function of normalized detuning $\Delta_1 = \delta_1/\alpha_1$. **f** Temporal profiles of the bright DQS at the FF, obtained by solving either the single mean-field equation (SMFE) or the coupled two-wave equation (CTWE), and at the SH, obtained from the CTWE. **g** Optical spectra of the bright DQS at the FF, obtained by solving either the SMFE or the CTWE, and at the SH, obtained from the CTWE. **h** Time domain propagation of the FF along the PPLN crystal. **i** Time domain propagation of the SH along the PPLN crystal. **j** Frequency domain propagation of the FF along the PPLN crystal. **k** Frequency domain propagation of the SH along the PPLN crystal.

changing the phase mismatch so that dissipative soliton generation is possible even with normal GVD.

Figures 2c, k illustrate a simulated example of nonlinearity-engineered DQS generation, where normal GVD is balanced by the negative EKN in a CW pumped free-space singly resonant cavity-enhanced second-harmonic generation system incorporating a PPLN bulk crystal. The simulation parameters are chosen to mirror the experimental setup illustrated in Fig. 1, including zero GVM and phase mismatch $\xi = -2\pi$. All the simulation parameters are listed in Supplementary Table 1. By scanning the pump frequency from the red to blue detuning side, the system sequentially transitions through states of CW, Turing pattern, chaos, and eventually reaches the soliton state at the blue detuning (Fig. 2c–2e), resembling conventional DKS dynamics. The temporal profiles and corresponding spectra of a pair of stable dual-color pulses at the output of the PPLN crystal are presented in Fig. 2f, g. The dual-color pulses are mutually trapped in the time domain, resulting in strong interaction. The FF spectrum follows a sech$^2$ profile, reflecting the balance between normal GVD and negative EKN, whereas the SH spectrum exhibits a flat-top profile with a weaker CW component. Further analysis reveals that back conversion is more evident in the SH than that observed in the FF (Fig. 2h, k). Consequently, the SH output spectrum is much more sensitive to phase mismatch that dictates the cascaded quadratic process and back conversion (see Supplementary Note 4).

Notably, the simulation results based on the coupled two-wave equation (CTWE, Supplementary Eqs. 1-4) provide a comprehensive description of the DQS dynamics, whereas those based on the SMFE (Eq. 1) offer an intuitive understanding of its underlying nature. The good agreement shown in Fig. 2f, g confirms the validity of the approximations used in deriving the SMFE. The slight discrepancy is attributed to the dispersion of the effective third-order nonlinearity (see Supplementary Note 1).

### Bright DQS generation

To implement DQS generation, we constructed a 387-MHz-FSR free-space cavity incorporating a z-cut-x-propagate bulk PPLN crystal (Fig. 1, see Methods). The quality (Q) factor at ~1571 nm was measured to be $2.136 \times 10^8$ for $p$ polarization. Importantly, only the FF is resonant within the cavity, while the non-resonant SH exits the cavity from mirror M2. According to the operating principle presented above, both the GVM and the phase mismatch are critical for DQS generation in the singly resonant cavity-enhanced second-harmonic generation. To achieve near-zero GVM across the bandwidth of erbium-doped fiber amplifier (EDFA), we employed type-I phase matching configuration ($o + o \rightarrow e$, nonlinear coefficient of $d_{eff} = 2.7$ pm/V) in the bulk PPLN crystal, where the zero-GVM wavelength is at ~1571 nm (see Supplementary Fig. 7). Notably, the GVDs at ~1571 nm and ~786 nm are both normal, with values of 105 fs$^2$/mm and 372 fs$^2$/mm, respectively, rendering conventional DKS generation impossible due to the positive MKN. The phase-matching condition was finely tuned by adjusting the PPLN crystal temperature, with a precision of 10 mK (see Methods and Supplementary Fig. 9). Additionally, to efficiently excite DQS at low average power, the cavity was synchronously pumped with a 12-ps pulse (see Methods and Supplementary Note 2).

As shown in Fig. 3a–d, we first investigated comb dynamics by scanning the FF frequency from red to blue detuning under conditions of zero GVM and $\xi = -2\pi$. During the scan, we observed transitions between Turing pattern (I), chaos (II) and soliton states (III), consistent with the simulations in Fig. 2c and S15. As predicted, the bright DQSs are generated at blue detuning, evident from clear soliton steps. The bright DQS optical spectra exhibit a sech$^2$ envelope at the FF and a flat-top envelope at the SH, with 3-dB bandwidths of 9 nm and 5 nm, respectively, in good agreement with the simulation results (Fig. 3e, f). The radio-frequency (RF) spectrum further confirms the high coherence and mutual trapping of the dual-color bright DQS. In addition,

Supplementary Fig. 12 shows the single sideband phase noise spectrum of the bright DQS. Finally, the FF pulse is nearly transform-limited, exhibiting a measured duration of 290 fs as retrieved from the intensity autocorrelation measurement (see Supplementary Fig. 13). This is the first experimental demonstration of bright DQS even in the normal dispersion regime.

The effect of GVM was studied by varying the FF wavelength to tune the GVM from 0 to 11 fs/mm. As shown in Figs. 3g–i, the asymmetry in the SH optical spectrum increases with larger GVM, and spectral fringes eventually appear, indicating SH pulse splitting caused by temporal walk-off during cascaded quadratic process and back conversion. These observations are in good agreement with the numerical results (see Supplementary Note 3). A slight asymmetry is also observed in the FF optical spectrum, though it is much less pronounced, as the high-Q FF resonance boundary condition suppresses the growth of such asymmetry.

The effect of phase mismatch was explored by adjusting the PPLN crystal temperature to change the phase mismatch $\xi$ from -1.4π to -2.4π. As shown in Fig. 3j, l, the SH optical spectrum is much more sensitive to phase mismatch, in good agreement with the numerical results (see Supplementary Note 4). While the FF optical spectra consistently maintain a sech$^2$ profile, the SH optical spectra exhibit greater variation—transitioning from a flat-top shape to a sech$^2$ profile and, in some cases, showing multiple fringes.

### Platicon generation

We then reversed the sign of the EKN to be positive—matching that of the MKN—in the normal-GVD PPLN crystal by adjusting the crystal temperature such that $\xi = 2\pi$. Under this condition, bistability exists between a low-intensity and a high-intensity homogeneous steady state (HSS). Due to the locking point (or Maxwell point) on the rising and falling edges of the pump pulse, platicons thus spontaneously emerge from stationary switching waves that connect the two steady state HSSs[33]. As shown in Fig. 4a–d, by scanning the FF frequency from red to blue detuning, we observed the generation of platicon at the red detuning. The platicon optical spectra exhibit a characteristic two-shoulder envelope at both the FF and SH. Increasing the red detuning further broadens the spectral bandwidth, consistent with numerical simulations and primarily attributed to the detuning-dependent shift of the Maxwell point[33]. The RF spectrum further confirms the high coherence and mutual trapping of the dual-color platicon. In addition, Supplementary Fig. 12 shows the single sideband phase noise spectrum of the platicon. Finally, a flat-top temporal profile, indicative of the platicon regime, was inferred from the measured triangular intensity autocorrelation trace (see Supplementary Fig. 13).

The effects of GVM and phase mismatch were examined and summarized in Fig. 4e–j. Similar to the behavior observed in bright DQS, GVM gives rise to asymmetry and fringes in the SH optical spectrum, whereas phase mismatch primarily modulates the bandwidth and overall spectral shape. Owing to the high peak power of the platicon, a green frequency comb was also observed, originating from SFG between the dual-color platicon pair within the PPLN crystal (see Supplementary Note 8).

### Discussion

In conclusion, we have provided a deeper theoretical understanding of DQS dynamics through nonlinearity engineering and successfully demonstrated DQS in a free-space, singly resonant, cavity-enhanced second-harmonic generation setup. Multi-color DQS combs through efficient intracavity nonlinear frequency conversion have also been observed. Furthermore, we have achieved in situ control over DQS comb properties and switching between bright DQSs and platicons, each with distinct spectro-temporal characteristics. Our work establishes DQS as a flexible light source that expands the reach of soliton-based technologies across a wide range of challenging wavelengths

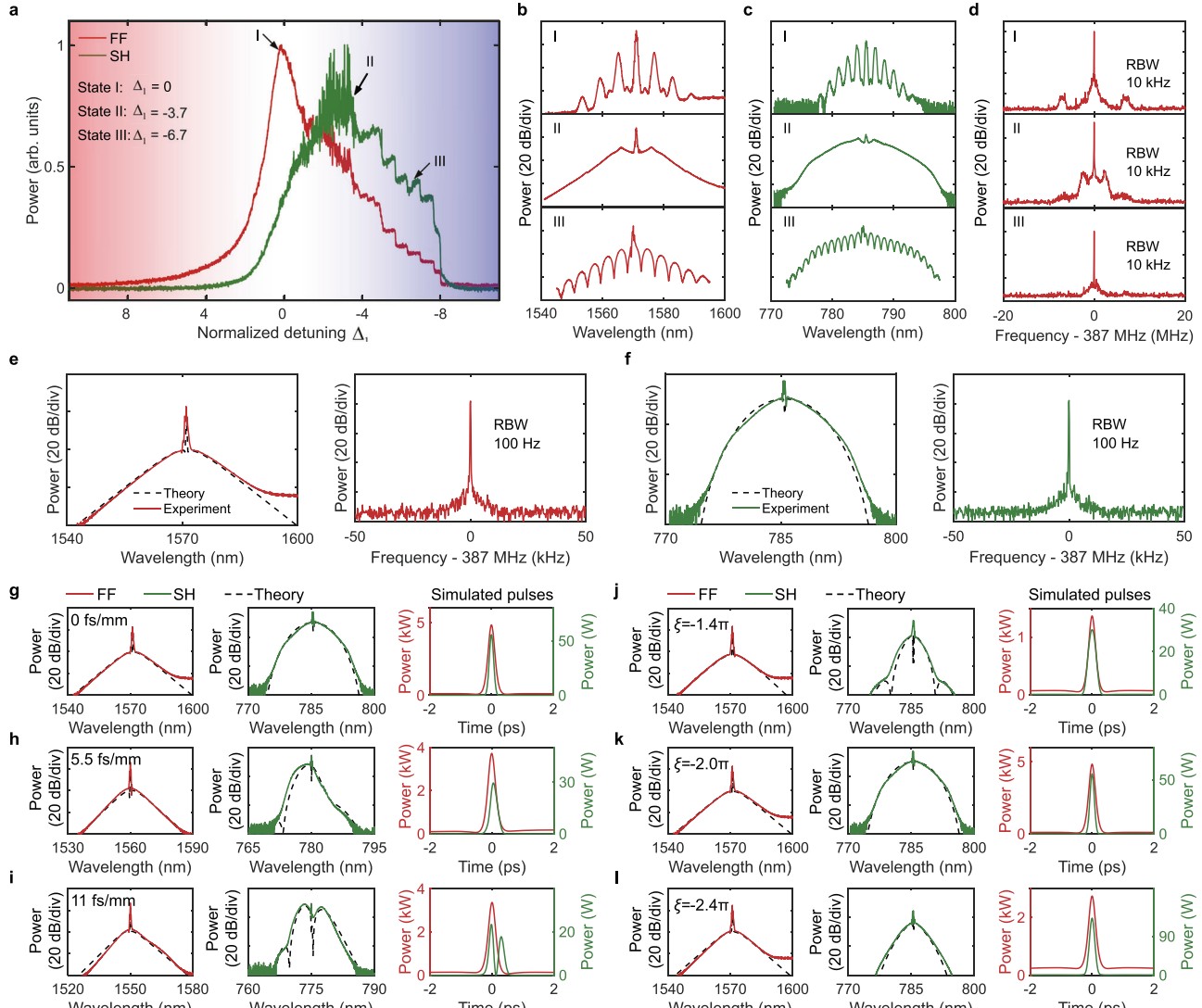

**Fig. 3 | Experimental generation of bright DQS. a** Output average power of the FF and SH as a function of normalized detuning $\Delta_1 = \delta_1/\alpha_1$. **b** Optical spectra of the bright DQS at the FF with different normalized detunings. **c** Optical spectra of the bright DQS at the SH with different normalized detunings. **d** RF spectra of the bright DQS at the FF with different detunings. **e** Optical spectrum (left) and RF spectrum (right) of the single bright DQS at the FF, taken with a normalized detuning of −7.6. **f** Optical spectrum (left) and RF spectrum (right) of the single bright DQS at the SH, taken with a normalized detuning of -7.6. The spectral shoulder beyond 1580 nm

originates from the amplified spontaneous emission of the L-band erbium-doped fiber amplifier. The identical repetition rates of the FF and SH confirm the mutual trapping of the dual-color bright DQS. **g–i** The effect of GVM on the optical spectrum at the FF and SH as well as the corresponding simulated temporal profiles. **g** GVM of 0 fs/mm and $\Delta_1$ of -7.6; (**h**) GVM of 5.5 fs/mm and $\Delta_1$ of −7.0; **i** GVM of 11 fs/mm and $\Delta_1$ of −6.8. **j–l** The effect of phase mismatch on the optical spectrum at the FF and SH. The third column shows the corresponding simulated temporal profiles. **j** $\xi$ of −1.4π and $\Delta_1$ of −6.4; (**k**) $\xi$ of −2.0π and $\Delta_1$ of -7.6; (**l**) $\xi$ of −2.4π and $\Delta_1$ of -7.6.

that are otherwise inaccessible. In addition, DQS can be seamlessly extended to emerging on-chip platforms such as thin-film PPLN[40], advancing the longstanding goal of integrated visible and ultraviolet mode-locked frequency combs. These advancements pave the way for transformative applications in fields such as optical atomic clocks[41], optical coherence tomography[42,43], astrocombs[44], and ultralow-threshold self-referenced frequency combs.

Our nonlinearity engineering approach introduces a new degree of freedom in cavity nonlinear optics, with implications far beyond ultrafast laser and frequency comb. For instance, by compensating the material Kerr nonlinearity, this method enables the maintenance of high intracavity power while suppressing modulation instability, enhancing the efficiency of entanglement sources in quantum photonic processing[45,46] and improving the performance of photonic sensors[47–49]. Moreover, the reconfigurable effective Kerr nonlinearity offers exciting opportunities for the development of innovative

photonic devices, including arbitrarily programmable nonlinear photonic circuits for artificial intelligence applications.

## Methods

### Experimental details

The free-space cavity is made from four-mirror bow-tie cavity with a FSR of -387 MHz. Three reflective mirrors (M1-M3) are coated with high reflectivity > 99.95% from 1530 nm to 1580 nm. The output coupler is coated with high reflectivity of -1.1% from 1530 nm to 1590 nm. The four mirrors are coated with high transmission > 95% from 760 nm to 790 nm. Both end facets of the 25-mm long 5% MgO-doped PPLN crystal are coated with high transmission at 1571 nm and 786 nm (reflectivity of 0.13% at 1571 nm and 0.09% at 786 nm). The cavity linewidth and Q factor are measured through a frequency-calibrated MZI with a FSR of 0.998 MHz. The group delay dispersion induced by the mirror coatings can be neglected. The poling period of the

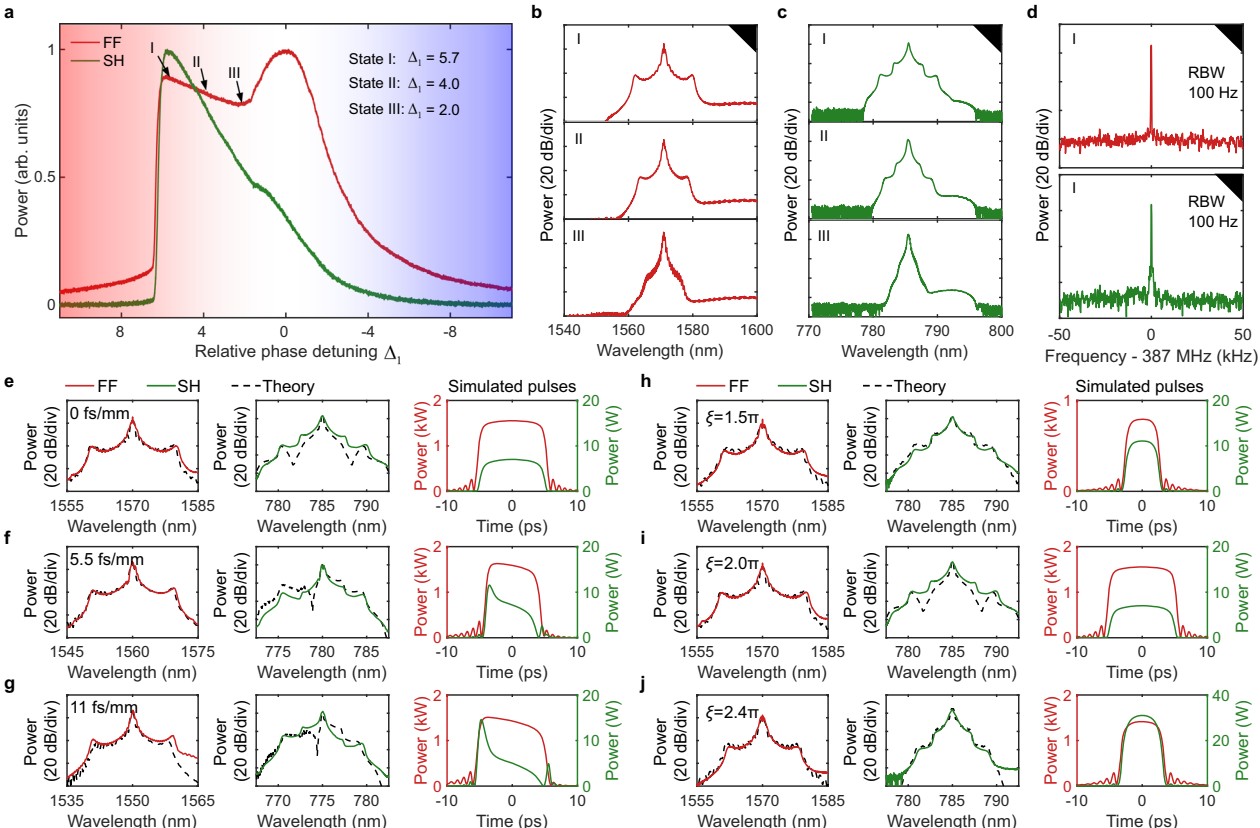

**Fig. 4 | Experimental generation of platicon. a** Output average power of the FF and SH as a function of normalized detuning $\Delta_1 = \delta_1/\alpha_1$. **b** Optical spectra of the platicon at the FF with different normalized detunings. **c** Optical spectra of the platicon at the SH with different normalized detunings. **d** RF spectra of the platicon at the FF (top) and SH (bottom) in state I with a normalized detuning of 5.7. The identical repetition rates of the FF and SH confirm the mutual trapping of the dual-color platicon. **e**–**g** The effect of GVM on the optical spectrum at the FF and SH as well as the corresponding simulated temporal profiles. **e** GVM of 0 fs/mm and $\Delta_1$ of 7.0; **f** GVM of 5.5 fs/mm and $\Delta_1$ of 6.8; (**g**) GVM of 11 fs/mm and $\Delta_1$ of 6.2. **h**–**j** The effect of phase mismatch on the optical spectrum at the FF and SH. The third column shows the corresponding simulated temporal profiles. **h** $\xi$ of 1.5π and $\Delta_1$ of 6.6; (**i**) $\xi$ of 2.0π and $\Delta_1$ of 7.0; (**j**) $\xi$ of 2.4π and $\Delta_1$ of 5.5.

intracavity PPLN crystal is 20.6 μm, which is designed for type-I phase matching ($o + o \rightarrow e$) at ~1571 nm. The PPLN crystal is temperature controlled with a resolution of 10 mK and the whole cavity is enclosed in a plastic box to prevent heat exchange with the air outside.

The pulsed pump laser is a home-built electro-optic frequency comb that generates a 12-ps dechirped Gaussian pulse train, amplified by a L-Band EDFA, with a repetition rate synchronized to the cavity FSR (see Supplementary Note 2). Of note, CW component coexists with the pulsed pump due to the insufficient extinction ratio of the pulse picking, which can explain the transmission differences between Fig. 2c and Fig. 3a (see Supplementary Fig. 15). The pump peak power is estimated to be 20 W, which is about 100 times lower than the threshold power of conventional DKS generation based on MKN in the same experimental configuration (Fig. 2b). To precisely control the detuning, a frequency-shifted (~1 FSR frequency shift from the pump) CW auxiliary laser is employed and locked to the free-space cavity via the Pound–Drever–Hall (PDH) locking technique. The PDH phase modulation frequency is 1 MHz, and an 80 kHz low-pass filter is used during PDH error signal demodulation. The CW auxiliary laser power is set to be 100 mW to prevent comb generation, and its polarization is identical with the pulsed pump ($p$ polarization). The pulsed pump laser and the CW auxiliary laser are counter propagated to minimize crosstalk and improve separation between the two beams.

The temperature for perfect phase matching condition at 1571 nm is obtained by locking a low-power (100 mW) CW laser to the cavity and monitoring the output SH laser power. Then the phase mismatch $\xi$ can be inferred according to the perfect phase matching temperature

point and the temperature-dependent phase mismatch diagram (see Supplementary Fig. 9).

## Data availability
All data generated or analyzed during this study are available within the paper and its Supplementary Information. Further source data will be made available on request.

## Code availability
The analysis codes will be available on request.

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

## Acknowledgements

The authors thank Scott Diddams from University of Colorado Boulder for lending the phase modulators and the fruitful suggestions. The authors thank the group member Jan Bartos for the help of building the pulsed pump laser. Mingming Nie (M.N.), Jonathan Musgrave (J.M.) and Shu-Wei Huang (S.W.H.) acknowledge the support from the National Science Foundation (ECCS2048202) and Office of Naval Research (N00014-22-1-2224).

## Author contributions

Mingming Nie (M.N.) and Shu-Wei Huang (S.W.H.) conceived the idea. M.N. and Jonathan Musgrave (J.M.) designed the experiment and con-ducted the theoretical framework and simulations. M.N. and J.M. per-formed the experiment. M.N., J.M., and S.W.H. conducted the data analysis. All authors contributed to the writing of the manuscript. S.W.H. supervised the whole project.

## Competing interests

Mingming Nie (M.N.), Jonathan Musgrave (J.M.), and Shu-Wei Huang (S.W.H.) are the inventors of a provisional patent application, filed by the University of Colorado Boulder, about the quadratic frequency combs approach.
