## [Transparent Peer Review file · Nature Communications]

Dissipative quadratic soliton in the cascaded nonlinearity limit

Corresponding Author: Professor Shu-Wei Huang

Version 0:

Reviewer comments:

Reviewer #1

(Remarks to the Author)

This paper reports on the generation of coherently driven soliton in the cascaded limit. It is a very nice first experimental demonstration, that I expect will attract significant attention.

The ability to excite bright solitons in normal dispersion regime is very promising and the fact that the sign on the nonlinearity can be easily tuned is very exciting

But I'm a bit surprised by some of the claims and lack of details or discussion on some aspects

First, I'm confused by the nomenclature used in the introduction.

CSs and DKSs are the same object (used by different communities), they are both the sech shape solution of the LLE, while here CSs is only introduced in the context of quadratic nonlinearities.

I would choose one name for Kerr solitons (and mention the other) and then explain that the soliton measured here is the cascaded version.

The way it is done here, the DKS connection is lost. I think it would make the main claim stronger.

I think that the authors should give more details about the experiment and soliton parameters in the main text. The pump power and the soliton duration for example.

There is good agreement with simulations, why not show the simulated temporal traces in the main text?

An important novelty with respect to standard CSs is the SH spectrum. It is shown and compared to simulations but there is little discussion on its shape. Beyond one sentence and a reference to the SM. A discussion on the temporal SH trace would be welcome to help understand the physics at play. It is not clear to me why the SH is split in two pulses and not simply broadened in time.

I think it's important because the authors claim in the abstract that synthetic four wave mixing is poorly understood but don't provide much information to help the reader understand. Especially for a journal with a broad audience.

Actually I don't agree with that claim. Cascaded nonlinearities are relatively well understood in the community, even in the coherently driven case, as discusses in the introduction.

I actually believe that it is not widespread because zero GVM is typically not easy to achieve. I'm a bit surprised by the zero GVM point at 1570 in a PPLN crystal. Is it commercially available ? Very few details are given. It would be nice to show some plots of the mismatch and its slope as a function of the wavelength.

I don't understand this sentence "However, this walk-off causes only slight asymmetry in the FF spectrum, primarily due to the intracavity self-reproduction condition " What is meant here?

Also some statements are a bit vague. For example the sentence "The pulse peak power is estimated to be 20 W, which is far below the threshold of conventional DKS generation based on MKN in the same experimental configuration. " is very qualitative. Why not calculate the required power based on the intrinsic χ^3 of the crystal? Based on figure 2a, I guess it's 100 times more?

I think one must be careful with the claim that it the first observation of the 1997 paper. It was a spatial soliton, which is the same mathematically of course (because of no GVM here)

but it was a doubly resonant configuration. The general concept of cascading is the same but the doubly resonant nonlinear response is different. The role of the phase mismatch is played by the pump detuning and the effective Kerr profile (as a function of the detuning) is quite different from the one shown here (fig 2a).

I think that this should be clarified.

I understand the spectral tails from desynchronization as shown in the SM but is that the reason for the tails around 1600 in fig 3b? If it's the case I would suggest to mention it in the main text.

Why does that spectrum stop around 1595?

There is little information about the pump pulses. It would be useful to add a spectrum, and maybe an autocorrelation trace. Any chirp might have an impact on the dynamics. Although the agreement is very convincing as is.

What is the conversion efficiency? In the singly resonant case, and when the GVM is low, it is probably not very efficient since most of the pump is not seen by the soliton. A signal to pump power ratio should be given. It would be nice to compare it to standard pulsed driven CSs. The authors say it is an efficient light source in the discussion, but the basis for this claim should be clarified.

Reviewer #2

(Remarks to the Author)

The manuscript presents an experimental demonstration of the generation of purely quadratic dissipative cavity solitons in a cavity-enhanced second-harmonic generation (SHG) configuration. The formation of frequency combs in quadratic resonators has garnered significant interest due to the development of highly efficient bulk and integrated nonlinear platforms, the potential for high effective Kerr nonlinearity, and the ability to access new spectral regions—particularly in the near-infrared, where conventional sources are limited. However, to date, the focus has mainly been on optical parametric oscillators (OPOs), where a continuous-wave or pulsed pump is converted into an optical frequency comb at lower frequencies. The originality of this work lies in its use of cascaded second-order nonlinearity in a singly resonant SHG configuration to achieve a large negative or positive effective Kerr nonlinearity to compensate the group velocity dispersion. This approach builds on the authors' previous theoretical studies.

The manuscript reports the experimental observation of bright quadratic solitons in the normal dispersion regime, both in the zero and small walk-off limits. Additionally, the formation of optical fronts is demonstrated in regimes with positive effective Kerr nonlinearity.

This study advances the understanding of cavity solitons in quadratic resonators and is of interest for the broader fields of nonlinear optics and optical frequency combs. The experimental findings are supported by numerical simulations, which appear to validate the interpretations presented.

Overall, I believe this work has the potential to be suitable for publication in Nature Communications. However, while the work seems solid, revisions are necessary to meet high scientific standards.

Specific comments and suggestions.

1/ Essential data and information are missing, preventing the reader from reproducing the experiments or numerical simulations, and hindering a proper comparison between them.

All simulation parameters must be provided. For example, in the simulations shown in Fig. 2, please specify the second-order nonlinear coefficient (κ), detuning, loss, dispersion, pump power, and the temporal shape of the pump (e.g., continuous wave, square pulse, Gaussian, etc.). The x-axis label in Fig. 2b should be "detuning" rather than "slow time," as detuning is the physically meaningful parameter. Additionally, the normalization in "arbitrary units" should be consistent for both the fundamental frequency (FF) and second harmonic (SH) powers within the same figure. The SH curve could be scaled by a fixed factor to avoid using multiple vertical axes.

The same comments apply to Fig. 3.

Regarding the experiments, more details are needed about the "custom electro-optic 12 ps pulse sources." What is the temporal shape of the pulse (e.g., Gaussian, flat-top, symmetric)? Is the pulse chirped? The spectrum of the pump should be provided, along with a characterization of the pulse.

In Fig. 3, the detuning value should be specified for all experiments. (Here, I refer to the phase detuning, not the pump frequency detuning, as the former is used in Eq. 1 and in the equations in the Supplementary Information.)

Please include the detuning value on the x-axis of Fig. 3a to enable comparison with the simulations.

Are all experiments conducted at the same pump power?

The same remarks apply to Fig. 4.

2/ What is the impact of the temporal shape of the pump? Are the QCS generated at the top of the 12ps pulse or locked in the pulse edge?

Furthermore, how does this behavior compare to that of standard Kerr cavity solitons under pulsed driving, as discussed in, for example, I. Hendry et al., Phys. Rev. A 97, 053834 (2018)? A discussion on whether the dynamics observed here differ from or resemble those in Kerr systems would strengthen the manuscript.

3/ The presence of narrow satellite peaks in the RF spectra is attributed by the authors to beating with the control signal.

However, the structure of the spectrum as observed in Fig. 3aIII resembles that of breathing solitons. The authors should provide a more detailed discussion to rule out the possibility that these features arise from oscillations of the QCSs. Are the amplitude of those RF peaks detuning dependent? Can these oscillations reproduced in simulations?

4/ While the experimental and simulated spectra generally agree, the experimental results occasionally exhibit spectral shoulders on either the red or blue side (see, for example, Fig. 3cl, Fig. 3d, Fig. 4a, and Fig. S17a & b). These shoulders extend well beyond the main QCS spectrum, are not reproduced in the simulations, and do not resemble typical dispersive waves.

The authors should comment on the possible origin of these features. A brief discussion in the manuscript would help clarify this point.

5/ In Fig3aIII, the spectrum shows the formation of bound states or soliton molecule. Is the formation mechanism with cascaded second order interaction identical to Kerr cavity solitons? The author should comment on this mechanism. Along the same line, is the SH pulse breakup as seen in FigS5 prevents the formation of bound-states?

6/ With positive effective loss, the authors claim they observe the formation of “dark DQCS”. In a previous publication (M. Nie, S.-W. Huang PRAp 13 044046 (2020)), the authors properly defined dark solitons: “Dark-quadratic-soliton pairs are formed from the locking of two fronts connecting two stable nontrivial cw solutions with identical amplitude but π phase difference.”

From the simulations in FigS18, I do not see any “dark solitons”, but nonlinear fronts which are not locked together to form dark pulses, least with a π -phase shift between them.

The author should thus not refer in the manuscript to the localized structure they observe as “dark solitons”, since the locking mechanism is not between two fronts but instead a pinning effect on the edges of the pulse pump, as explained in the manuscript and well-known in standard Kerr cavity under synchronous pulse driving with opposite sign for the Kerr nonlinearity and the dispersion coefficient.

7/ The discussion of the difference between the RF single-sideband phase noise (SSBPN) of the bright QCSs and fronts, and the comparison with the pump SSBPN, is rather vague. The results suggest that bright QCSs are more prone to noise than fronts and increase the pump noise. However, we would expect the reverse, since cavity solitons, as dissipative structures, should filter out the noise of the pump—at least in some frequency bands—while Fig. S11 shows a much larger SSBPN, which is unusual.

8/ In the perspective of generating two-colour combs via CE-SHG, and beyond the lower power threshold and the control of the sign of the effective third order nonlinearity, are there advantages of generating QCS compared to separating the Kerr soliton formation and the intracavity CE-SHG (bandwidth, efficiency, coherence, etc.)?

9/ The introduction should be revised, to be more appealing and easier to follow, specifically the structure. Dissipative solitons were first observed in fiber cavities, this should be properly acknowledged, all the more given the “historical” perspective of the introduction. The recent work of Tang (Opt Lett 49, 2449 (2024)) closely related with the present work should also be cited.

10/ The quality of the figures is poor and looks like assembled screenshots. All figures should be labelled. The axis label should be consistent e.g. not mix “Bandwidth” and “Wavelength”. All curves should be discussed in the caption. CTWE, SMFE should be defined or referred to equations in the main manuscript or SI.

11/ Other comments:

- A lot of acronyms are used thorough the paper, making it quite difficult to follow.
- There is a confusion about (ξ) and the wave-vector mismatch parameter. In the manuscript, it is said: “ $\xi = \Delta k_L$ is the wave-vector mismatch parameter”; or “When the wave-vector mismatch $\xi = m \cdot 2\pi$ ”; but also “ Δk is the wave-vector mismatch”. ξ is not a wave vector mismatch. Please revise for correctness and consistency.
- Please do not use the term “soliton pair” when referring to the FF-SH locked dissipative structures, given the conventional meaning of a soliton pair in the context of dissipative solitons. Moreover, the SH pulse does not exhibit solitonic behavior and should not be referred to as a soliton.
- In Fig1. The word “Length” is not appropriate. In the bottom right inset, I suggest replacing “Large and negative NL” by “effective negative Kerr nonlinearity”, which is more specific to this work.
- The first sentence of the Discussion section should be revised.

Reviewer #3

(Remarks to the Author)

The authors studied dissipative quadratic cavity solitons in free space and with the infrared pump beam resonance only. This experimental study also relies on a unique mean-field equation, derived to describe dissipative quadratic cavity solitons and extract the dynamics of the resonant beam, thus producing a pair of two-color pulses. Phase-mismatching conditions allow switching between bright and dark solitary waves, each exhibiting distinct spectrotemporal characteristics. Thus, in addition to SH combs centered near the second harmonic position, other visible combs centered at lower wavelengths are also generated by intrinsic sum frequency generation.

The work shown in that paper is of significance in the field of quadratic soliton cavity and the results are good and convincing.

The experimental demonstration and theoretical approach are well presented and are in good agreement. The methodology is good and the work is sufficiently well described.

The article is of good quality.

Remark : The first theoretical published paper on quadratic solitons is not the one referenced in [17] but the one published by Karamzin and Sukhorukov in 1974 (Theoretical prediction). The authors could add this reference "Karamzin, Y. N., & Sukhorukov, A. P. (1974). Nonlinear interaction of diffracted light beams in a medium with quadratic nonlinearity: mutual focusing of beams and limitation on the efficiency of optical frequency converters. *ZhETF Pisma Redaktsiiu*, 20, 734."

Q1: Could the authors explain why they used a PPLN in a type 1 phase matching and not a type 0 phase matching?

Q2: Could the authors explain why they don't try to make the second harmonic resonant?

Q3: The shape of solitons in the time domain is not presented in the main article. It would be interesting to present both the spectral and temporal evolutions in the same figure

Q4: Nonlinear conversion affects the time domain but can also introduce spatial distortion. Did the authors observe any trapping or spatial broadening of the beam?

Version 1:

Reviewer comments:

Reviewer #1

(Remarks to the Author)

The paper has been improved, my comments have been taken into account. I support publication of the manuscript as is.

Reviewer #2

(Remarks to the Author)

The authors have satisfactorily addressed most of the reviewer's concerns. However, though the results are of interest and deserved to be published, I still have several comments that prevent me from supporting the publication of the manuscript in its current form.

1/ In their revised version, the authors continue to employ terminology that is inappropriate for describing the dissipative structure observed in the normal dispersion regime (see point 7 of Rev2 comments). True dark pulses with synchronous pumping in normal dispersion Kerr cavities have been reported with a clear dip in the pulse center (see Garbin, Experimental and numerical investigations of switching wave dynamics in a normally dispersive fibre ring resonator 2017). On the contrary, Fig4 of the manuscript shows the formation of BRIGTH pulses, which cannot thus be called "dark pulses". The authors must change how they called this dissipative structure.

2/ In the revised introduction (and in Supplemental Information section VI), the first sentence is not correct. The loss experienced by the soliton is balanced by the coherent driving, not by "parametric gain" in dispersive Kerr resonators. Note that parametrically driven Kerr cavity solitons, which are sustained by the equilibrium between parametric (phase sensitive) gain and loss, also exist and represent distinct types of solitons.

3/ The authors explain in the revised caption of Figure3 that the asymmetry is due to the amplified spontaneous emission in EDFA. The existence of an EDFA in the experiment must be mentioned in the Method section and in the Supp Info Section about the pumped laser (not just mentioned in the caption of a figure).

4/ Fig5c: the labels of the axes are not appropriate: horizontal is "delay" and vertical is "Autocorrelation signal".

5/ The answer about the SSB phase noise does not convince me. Experiments are reported in the literature about (long) synchronously pumped cavities without repetition rate active stabilization. It is expected the soliton, as a dissipative structure in a cavity to filters the jitter of the noise and thus the comb exhibits lower SSB phase noise. One explanation could be a jitter induced by the noise injected by the ASE of the amplifier in the pump laser source.

Reviewer #3

(Remarks to the Author)

Comments and questions I addressed to the authors have been properly taken into account by the authors, and I have no further requests regarding this manuscript.

Reviewer #1 (Remarks to the Author):

This paper reports on the generation of coherently driven soliton in the cascaded limit. It is a very nice first experimental demonstration, that I expect will attract significant attention.

The ability to excite bright solitons in normal dispersion regime is very promising and the fact that the sign on the nonlinearity can be easily tuned is very exciting.

But I'm a bit surprised by some of the claims and lack of details or discussion on some aspects

First, I'm confused by the nomenclature used in the introduction.

CSs and DKSs are the same object (used by different communities), they are both the sech shape solution of the LLE, while here CSs is only introduced in the context of quadratic nonlinearities.

I would choose one name for Kerr solitons (and mention the other) and then explain that the soliton measured here is the cascaded version.

The way it is done here, the DKS connection is lost. I think it would make the main claim stronger.

Response: Following the reviewer's suggestion, we have revised the Introduction to use the terms DKS and DQS consistently, removing CS to avoid confusion and strengthen the connection between DKS and DQS. We have also rewritten the Introduction to make it more concise and engaging, in response to the other reviewers' comments.

I think that the authors should give more details about the experiment and soliton parameters in the main text. The pump power and the soliton duration for example.

Response: Following the reviewer's suggestion, we have provided more details about the experiment and soliton parameters in the revised manuscript.

Revisions made: Besides the pump information already given in Methods, we have added a new Supplementary Information Section II that provides the schematic and characterization of the pump. For the DQS pulse duration, we have added a new Supplementary Information Section VII that presents the intensity autocorrelation traces of both the bright DQS and dark pulse.

II. Experimental details of the pulsed pump laser

The pulsed pump laser is a home-built electro-optic frequency comb consisting of one intensity modulator and two phase modulators driven by a 3.1-GHz RF source (Fig. S5a). A 1/8 pulse picker is then used to reduce the repetition rate to 387 MHz that matches the DQS cavity FSR. Fig. S5b shows the comb spectra before and after the pulse picker, and Fig. S5c shows the intensity autocorrelation of the 12-ps dechirped Gaussian pulse after the pulse picker.

Fig. S5. (a) Schematic of the pulsed pump laser. (b) Optical spectra of the pulsed pump laser before and after the pulse picker. (c) Intensity autocorrelation of the 12-ps dechirped Gaussian pulse after the pulse picker.

VII. Intensity autocorrelation of bright DQS and dark pulse

Figure S13a shows the optical spectrum of the single bright DQS after removing the pump background with a volume Bragg grating (VBG), and Fig. S13b presents the corresponding intensity autocorrelation trace, confirming its femtosecond pulse characteristics. Figure S13c shows the optical spectrum of the dark pulse in state I at a normalized detuning of 5.7, and Fig. S13d presents the corresponding intensity autocorrelation trace, confirming its flat-top temporal profile.

Fig. S13. (a) Optical spectrum of the single bright DQS with the pump background removed by VBG. (b) The corresponding intensity autocorrelation trace. (c) Optical spectrum of the dark pulse in state I with a normalized detuning of 5.7. (d) The corresponding intensity autocorrelation trace.

There is good agreement with simulations, why not show the simulated temporal traces in the main text?

Response: Following the reviewer's suggestion, we have included the simulated temporal traces in Figs. 3 and 4 of the main text. In addition, we have added a new Supplementary Information Section VII that presents the intensity autocorrelation traces of both the bright DQS and dark pulse.

Revisions made:

Fig. 3. Experimental generation of bright DQS. (a) Output average power of the FF and SH as a function of normalized detuning $\Delta_1 = \delta_1 / \alpha_1$. (b) Optical spectra of the bright DQS at the FF with different normalized detunings. (c) Optical spectra of the bright DQS at the SH with different normalized detunings. (d) RF spectra of the bright DQS at the FF with different detunings. (e) Optical spectrum (left) and RF spectrum (right) of the single bright DQS at the FF, taken with a normalized detuning of -7.6 . (f) Optical spectrum (left) and RF spectrum (right) of the single bright DQS at the SH, taken with a normalized detuning of -7.6 . The spectral shoulder beyond 1580 nm originates from the amplified spontaneous emission of the L-band erbium-doped fiber amplifier. The identical repetition rates of the FF and SH confirm the mutual trapping of the dual-color bright DQS. (g)-(i) The effect of GVM on the optical spectrum at the FF and SH. The third column shows the corresponding simulated temporal profiles. (g) GVM=0 fs/mm, $\Delta_1 = -7.6$; (h) GVM=5.5 fs/mm, $\Delta_1 = -7.0$; (i) GVM=11 fs/mm, $\Delta_1 = -6.8$. (j)-(l) The effect of phase mismatch on the optical spectrum at the FF and SH. The third column shows the corresponding simulated temporal profiles. (j) $\xi = -1.4\pi$, $\Delta_1 = -6.4$; (k) $\xi = -2.0\pi$, $\Delta_1 = -7.6$; (l) $\xi = -2.4\pi$, $\Delta_1 = -7.6$.

Fig. 4. Experimental generation of dark pulse. (a) Output average power of the FF and SH as a function of normalized detuning $\Delta_1 = \delta_1/\alpha_1$. (b) Optical spectra of the dark pulse at the FF with different normalized detunings. (c) Optical spectra of the dark pulse at the SH with different normalized detunings. (d) RF spectra of the dark pulse at the FF (top) and SH (bottom) in state I with a normalized detuning of 5.7. The identical repetition rates of the FF and SH confirm the mutual trapping of the dual-color dark pulse. (e)-(g) The effect of GVM on the optical spectrum at the FF and SH. The third column shows the corresponding simulated temporal profiles. (e) GVM=0 fs/mm, $\Delta_1=7.0$; (f) GVM=5.5 fs/mm, $\Delta_1=6.8$; (g) GVM=11 fs/mm, $\Delta_1=6.2$. (h)-(j) The effect of phase mismatch on the optical spectrum at the FF and SH. The third column shows the corresponding simulated temporal profiles. (h) $\zeta=1.5\pi$, $\Delta_1=6.6$; (i) $\zeta=2.0\pi$, $\Delta_1=7.0$; (j) $\zeta=2.4\pi$, $\Delta_1=5.5$.

VII. Intensity autocorrelation of bright DQS and dark pulse

Figure S13a shows the optical spectrum of the single bright DQS after removing the pump background with a volume Bragg grating (VBG), and Fig. S13b presents the corresponding intensity autocorrelation trace, confirming its femtosecond pulse characteristics. Figure S13c shows the optical spectrum of the dark pulse in state I at a normalized detuning of 5.7, and Fig. S13d presents the corresponding intensity autocorrelation trace, confirming its flat-top temporal profile.

Fig. S13. (a) Optical spectrum of the single bright DQS with the pump background removed by VBG. (b) The corresponding intensity autocorrelation trace. (c) Optical spectrum of the dark pulse in state I with a normalized detuning of 5.7. (d) The corresponding intensity autocorrelation trace.

An important novelty with respect to standard CSs is the SH spectrum. It is shown and compared to simulations but there is little discussion on its shape. Beyond one sentence and a reference to the SM. A discussion on the temporal SH trace would be welcome to help understand the physics at play. It is not clear to me why the SH is split in two pulses and not simply broadened in time. I think it's important because the authors claim in the abstract that synthetic four wave mixing is poorly understood but don't provide much information to help the reader understand. Especially for a journal with a broad audience. Actually I don't agree with that claim. Cascaded nonlinearities are relatively well understood in the community, even in the coherently driven case, as discussed in the introduction. I actually believe that it is not widespread because zero GVM is typically not easy to achieve.

Response: Following the reviewer's suggestion, we have removed the statement in the Abstract and Introduction claiming that synthetic four-wave mixing is poorly understood.

On the other hand, we want to emphasize that most previous studies on cascaded nonlinearities have focused on the large amplitude and sign reversal of the effective nonlinearity at the pump frequency when an intense pulsed laser propagates through a phase-mismatched quadratic crystal in a single pass. In contrast, the spectral bandwidth of the cascaded nonlinearity—particularly in a cavity configuration—has been far less explored, despite its central role in DQS formation. Our results reveal that the effective two-photon absorption (ETPA) and effective Kerr nonlinearity (EKN) depend strongly on the group-velocity mismatch (GVM). A small GVM leads to a broad bandwidth of both ETPA and EKN, thereby enabling DQS generation, as confirmed experimentally in our study.

The figure below shows the temporal evolution of the dual-color pulse within the PPLN crystal. The SH temporal trace splits into two pulses as a result of both group-velocity mismatch (GVM) and phase mismatch. At different propagation distances, the back-conversion rates between the FF and SH fields vary. Due to the temporal walk-off between the two fields, the newly generated SH pulse from the FF conversion becomes temporally shifted relative to the original SH pulse. Under

perfect phase matching and in the absence of back-conversion, the SH pulse would simply broaden in time owing to the walk-off, as the reviewer anticipated.

Revisions made: We have expanded the discussion about the GVM effect on SH in the **Bright DQS generation** section as follows: “The effect of GVM was studied by varying the FF wavelength to tune the GVM from 0 to 11 fs/mm. As shown in Figs. 3g–3i, the asymmetry in the SH optical spectrum increases with larger GVM, and spectral fringes eventually appear, indicating SH pulse splitting caused by temporal walk-off during cascaded quadratic process and back conversion. These observations are in good agreement with the numerical results (see Supplementary Information Section III). A slight asymmetry is also observed in the FF optical spectrum, though it is much less pronounced, as the high-Q FF resonance boundary condition suppresses the growth of such asymmetry.”

I’m a bit surprised by the zero GVM point at 1570 in a PPLN crystal. Is it commercially available? Very few details are given. It would be nice to show some plots of the mismatch and its slope as a function of the wavelength.

Response: Yes, we used a commercially available PPLN crystal to demonstrate the DQS. Information about the PPLN crystal was already included in the **Bright DQS generation** section as “To achieve near-zero GVM across the bandwidth of erbium-doped fiber amplifier (EDFA), we employed type-I phase matching configuration ($o+o\rightarrow e$, nonlinear coefficient of $d_{eff}=2.7$ pm/V) in the bulk PPLN crystal, where the zero-GVM wavelength is at ~ 1571 nm (see Fig. S7 in Supplementary Information Section III)”. More information about the PPLN crystal was given in the **Methods** section. Finally, Figure S7 plots the GVM as a function of FF wavelength at different PPLN crystal temperatures.

I don’t understand this sentence “However, this walk-off causes only slight asymmetry in the FF spectrum, primarily due to the intracavity self-reproduction condition “ What is meant here?

Response: To improve clarity, we have expanded the discussion as follows: “The effect of GVM was studied by varying the FF wavelength to tune the GVM from 0 to 11 fs/mm. As shown in Figs. 3g–3i, the asymmetry in the SH optical spectrum increases with larger GVM, and spectral fringes

eventually appear, indicating SH pulse splitting caused by temporal walk-off during cascaded quadratic process and back conversion. These observations are in good agreement with the numerical results (see Supplementary Information Section III). A slight asymmetry is also observed in the FF optical spectrum, though it is much less pronounced, as the high-Q FF resonance boundary condition suppresses the growth of such asymmetry.”

Also some statements are a bit vague. For example the sentence “The pulse peak power is estimated to be 20 W, which is far below the threshold of conventional DKS generation based on MKN in the same experimental configuration. “ is very qualitative. Why not calculate the required power based on the intrinsic χ^3 of the crystal? Based on figure 2a, I guess it's 100 times more?

Response: To be quantitative, we have revised the sentence as follows: “The pump peak power is estimated to be 20 W, which is about 100 times lower than the threshold power of conventional DKS generation based on MKN in the same experimental configuration (Fig. 2b).”

I think one must be careful with the claim that it is the first observation of the 1997 paper. It was a spatial soliton, which is the same mathematically of course (because of no GVM here), but it was a doubly resonant configuration. The general concept of cascading is the same but the doubly resonant nonlinear response is different. The role of the phase mismatch is played by the pump detuning and the effective Kerr profile (as a function of the detuning) is quite different from the one shown here (fig 2a).

I think that this should be clarified.

Response: To avoid confusion and improve clarity, we have removed the references and claims related to spatial solitons. We have also rewritten the Introduction to make it more concise and engaging, in response to the other reviewers’ comments.

I understand the spectral tails from desynchronization as shown in the SM but is that the reason for the tails around 1600 in fig 3b? If it’s the case I would suggest to mention it in the main text.

Why does that spectrum stop around 1595?

Response: The spectral tails actually come from the amplified spontaneous emission (ASE) of the L-band EDFA. We have added the following explanatory sentence to the caption of Fig. 3: “The spectral shoulder beyond 1580 nm originates from the amplified spontaneous emission of the L-band erbium-doped fiber amplifier.” In addition, we have replotted Fig. 3e to match the wavelength range of 1540–1600 nm used for the other optical spectra in Fig. 3.

There is little information about the pump pulses. It would be useful to add a spectrum, and maybe an autocorrelation trace. Any chirp might have an impact on the dynamics. Although the agreement is very convincing as is.

Response: Following the reviewer’s suggestion, we have provided more details about the pump.

Revisions made: Besides the pump information already given in Methods, we have added a new Supplementary Information Section II that provides the pump schematic and characterization.

II. Experimental details of the pulsed pump laser

The pulsed pump laser is a home-built electro-optic frequency comb consisting of one intensity modulator and two phase modulators driven by a 3.1-GHz RF source (Fig. S5a). A 1/8 pulse picker is then used to reduce the repetition rate to 387 MHz that matches the DQS cavity FSR. Fig. S5b shows the comb spectra before and after the pulse picker, and Fig. S5c shows the intensity autocorrelation of the 12-ps dechirped Gaussian pulse after the pulse picker.

Fig. S5. (a) Schematic of the pulsed pump laser. (b) Optical spectra of the pulsed pump laser before and after the pulse picker. (c) Intensity autocorrelation of the 12-ps dechirped Gaussian pulse after the pulse picker.

What is the conversion efficiency? In the singly resonant case, and when the GVM is low, it is probably not very efficient since most of the pump is not seen by the soliton. A signal to pump power ratio should be given. It would be nice to compare it to standard pulsed driven CSs. The authors say it is an efficient light source in the discussion, but the basis for this claim should be clarified.

Response: Since there exists a continuous-wave component in the pump (due to low extinction ratio during the pulse picking process), it is challenging to reliably determine the conversion efficiency experimentally. Thus, we relied on numerical simulations—which have been shown to

agree well with experiments—to quote the conversion efficiency. The estimated conversion efficiency is ~3.6%, comparable to that of standard pulse-driven DKS generation [Optica **9**, 231–239 (2022)], indicating a high degree of similarity between conventional DKS and DQS in the singly resonant CE-SHG system.

To avoid confusion, we have removed the use of “efficient light source” in the discussion. What we meant is that DQS enables efficient extension of the comb spectrum into unconventional wavelength bands that are inaccessible to DKS combs.

Reviewer #2 (Remarks to the Author):

The manuscript presents an experimental demonstration of the generation of purely quadratic dissipative cavity solitons in a cavity-enhanced second-harmonic generation (SHG) configuration. The formation of frequency combs in quadratic resonators has garnered significant interest due to the development of highly efficient bulk and integrated nonlinear platforms, the potential for high effective Kerr nonlinearity, and the ability to access new spectral regions—particularly in the near-infrared, where conventional sources are limited. However, to date, the focus has mainly been on optical parametric oscillators (OPOs), where a continuous-wave or pulsed pump is converted into an optical frequency comb at lower frequencies. The originality of this work lies in its use of cascaded second-order nonlinearity in a singly resonant SHG configuration to achieve a large negative or positive effective Kerr nonlinearity to compensate the group velocity dispersion. This approach builds on the authors' previous theoretical studies.

The manuscript reports the experimental observation of bright quadratic solitons in the normal dispersion regime, both in the zero and small walk-off limits. Additionally, the formation of optical fronts is demonstrated in regimes with positive effective Kerr nonlinearity.

This study advances the understanding of cavity solitons in quadratic resonators and is of interest for the broader fields of nonlinear optics and optical frequency combs. The experimental findings are supported by numerical simulations, which appear to validate the interpretations presented.

Overall, I believe this work has the potential to be suitable for publication in Nature Communications. However, while the work seems solid, revisions are necessary to meet high scientific standards.

Specific comments and suggestions.

1/ Essential data and information are missing, preventing the reader from reproducing the experiments or numerical simulations, and hindering a proper comparison between them.

All simulation parameters must be provided. For example, in the simulations shown in Fig. 2, please specify the second-order nonlinear coefficient (κ), detuning, loss, dispersion, pump power, and the temporal shape of the pump (e.g., continuous wave, square pulse, Gaussian, etc.). The x-axis label in Fig. 2b should be “detuning” rather than “slow time,” as detuning is the physically meaningful parameter. Additionally, the normalization in “arbitrary units” should be consistent for both the fundamental frequency (FF) and second harmonic (SH) powers within the same figure. The SH curve could be scaled by a fixed factor to avoid using multiple vertical axes.

The same comments apply to Fig. 3.

Response: Following the reviewer's suggestion, we have added a new Table S1 at the end of Supplementary Information Section I that includes all the simulation parameters used in the main text. As suggested, we have updated the x-axis label to “Normalized detuning” in the revised Figs. 2c–2e, 3a, 4a, and S15–S18. Finally, we have applied a scaling factor to the SH curves in all simulated figures (Figs. 2c and S15–S18).

Regarding the experiments, more details are needed about the “custom electro-optic 12 ps pulse sources.” What is the temporal shape of the pulse (e.g., Gaussian, flat-top, symmetric)? Is the pulse chirped? The spectrum of the pump should be provided, along with a characterization of the pulse.

In Fig. 3, the detuning value should be specified for all experiments. (Here, I refer to the phase detuning, not the pump frequency detuning, as the former is used in Eq. 1 and in the equations in the Supplementary Information.)

Please include the detuning value on the x-axis of Fig. 3a to enable comparison with the simulations.

Are all experiments conducted at the same pump power?

The same remarks apply to Fig. 4.

Response: Following the reviewer’s suggestion, we have provided more details about the pump. Furthermore, we have updated the x-axis label to “Normalized detuning” and indicated the corresponding normalized detuning values used in the experiments in the revised Figs. 3 and 4.

Yes, all the experiments are conducted at the same pump power.

Revisions made: Besides the pump information already given in Methods, we have added a new Supplementary Information Section II that provides the pump schematic and characterization.

III. Experimental details of the pulsed pump laser

The pulsed pump laser is a home-built electro-optic frequency comb consisting of one intensity modulator and two phase modulators driven by a 3.1-GHz RF source (Fig. S5a). A 1/8 pulse picker is then used to reduce the repetition rate to 387 MHz that matches the DQS cavity FSR. Fig. S5b shows the comb spectra before and after the pulse picker, and Fig. S5c shows the intensity autocorrelation of the 12-ps dechirped Gaussian pulse after the pulse picker.

Fig. S5. (a) Schematic of the pulsed pump laser. (b) Optical spectra of the pulsed pump laser before and after the pulse picker. (c) Intensity autocorrelation of the 12-ps dechirped Gaussian pulse after the pulse picker.

2/ What is the impact of the temporal shape of the pump? Are the QCS generated at the top of the 12ps pulse or locked in the pulse edge?

Furthermore, how does this behavior compare to that of standard Kerr cavity solitons under pulsed driving, as discussed in, for example, I. Hendry et al., Phys. Rev. A 97, 053834 (2018)? A discussion on whether the dynamics observed here differ from or resemble those in Kerr systems would strengthen the manuscript.

Response: We appreciate the reviewer's insightful questions. We have added new Supplementary Information Sections XII and XIII, which present numerical studies on (i) the effect of pump pulse shape on the bright DQS and dark pulse, and (ii) the trapping dynamics of the bright DQS by the pulsed pump, respectively.

Revisions made: The effect of pump pulse shape is numerically studied and presented in Supplementary Information Section XII. The trapping dynamics of the bright DQS is numerically studied and presented in Supplementary Information Section XIII.

XII. Effect of pump pulse shape on bright DQS and dark pulse

Fig. S22 shows the effect of pump pulse shape on the bright DQS and dark pulse. While the pump shape has minimal impact on the spectral and temporal characteristics of the bright DQS, its effect on the dark pulse is more pronounced, reflecting their distinct formation mechanisms.

Fig. S22. (a) Bright DQS generation with a Gaussian pulse. (b) Bright DQS generation with a flat-top pulse. (c) Dark pulse generation with a Gaussian pulse. (d) Dark pulse generation with a flat-top pulse. The first column shows the temporal profiles of the pump pulses, the second column shows the temporal profiles of the generated pulses, and the third column shows their corresponding optical spectra. The pump pulse duration is fixed at 12 ps for all cases.

XIII. Trapping of bright DQS by pulsed pump

To investigate the DQS trapping dynamics, we numerically solve the CTWE (Eqs. S1-S4) using an initial condition comprising a short perturbation approximating a DQS that is offset from the pulsed pump peak. This perturbation evolves into a DQS that may drift under the influence of the pump's amplitude gradient. The simulation was continued until a steady state was reached. Figs. S23 summarizes the results. The DQSs are generally attracted toward the edge of the pulsed pump, with their equilibrium positions dependent on the pump power. This trapping behavior closely resembles that observed in conventional pulse-driven DKSS (Fig. S24) [7].

Fig. S23. (a) Steady-state DQS temporal profiles, (b) temporal evolution of the FF, and (c) temporal evolution of the SH, obtained for a pump peak power of 15 W. (d) Steady-state DQS temporal profiles, (e) temporal evolution of the FF, and (f) temporal evolution of the SH, obtained for a pump peak power of 25 W. (g) Steady-state DQS temporal profiles, (h) temporal evolution of the FF, and (i) temporal evolution of the SH, obtained for a pump peak power of 35 W.

Fig. S24. (a) Steady-state DKS temporal profiles, and (b) temporal evolution of the DKS, obtained for a pump peak power of 7 W. (c) Steady-state DKS temporal profiles, and (d) temporal evolution of the DKS, obtained for a pump peak power of 10 W. (e) Steady-state DKS temporal profiles, and (f) temporal evolution of the DKS, obtained for a pump peak power of 15 W.

3/ The presence of narrow satellite peaks in the RF spectra is attributed by the authors to beating with the control signal. However, the structure of the spectrum as observed in Fig. 3aIII resembles that of breathing solitons. The authors should provide a more detailed discussion to rule out the possibility that these features arise from oscillations of the QCSs. Are the amplitude of those RF peaks detuning dependent? Can these oscillations reproduced in simulations?

Response: We appreciate the reviewer’s insightful questions. To completely remove any ambiguity, we refined the experimental setup to suppress backscattering of the control signal. In the revised Fig. 3d, no satellite peaks are observed, confirming the absence of breather solitons.

4/ While the experimental and simulated spectra generally agree, the experimental results occasionally exhibit spectral shoulders on either the red or blue side (see, for example, Fig. 3cI, Fig. 3d, Fig. 4a, and Fig. S17a & b). These shoulders extend well beyond the main QCS spectrum, are not reproduced in the simulations, and do not resemble typical dispersive waves.

The authors should comment on the possible origin of these features. A brief discussion in the manuscript would help clarify this point.

Response: In the experiment, we used either C-band (1530-1565 nm) or L-band (1570-1605 nm) to amplify the ~1570 nm pump laser. Since the wavelength 1570 nm is at the edge of the amplifying bandwidth of the C-band or L-band EDFAs, there will be strong amplified spontaneous emission (ASE) at the blue side of the C-band EDFA output or at the red side of the L-band EDFA output. Therefore, the spectral tails in the spectra of bright soliton or locked fronts originate from the ASE. We have added the following explanatory sentence to the caption of Fig. 3: “The spectral shoulder beyond 1580 nm originates from the amplified spontaneous emission of the L-band erbium-doped fiber amplifier.”

5/ In Fig3aIII, the spectrum shows the formation of bound states or soliton molecule. Is the formation mechanism with cascaded second order interaction identical to Kerr cavity solitons? The author should comment on this mechanism. Along the same line, is the SH pulse breakup as seen in FigS5 prevents the formation of bound-states?

Response: We think that the formation mechanism of bright DQS molecules is identical to DKS: long-range soliton interactions mediated by resonator mode degeneracies from avoided mode crossing [Nat. Photon. **11**, 671 (2017)]. There are orthogonal polarized mode families with different spatial modes in our free-space cavity that can cause avoided mode crossings.

The SH pulse breakup in revised Fig. S6 (original Fig. S5) is caused by the group velocity mismatch (GVM) between FF and SH. With GVM of 11 fs/mm, we also observe the formation of soliton molecules (see the Figure on the right).

Therefore, the SH pulse breakup will not prevent the formation of bound states or soliton molecules.

6/ With positive effective loss, the authors claim they observe the formation of “dark DQS”. In a previous publication (M. Nie, S.-W. Huang PRAp 13 044046 (2020)), the authors properly defined dark solitons: “Dark-quadratic-soliton pairs are formed from the locking of two fronts connecting two stable nontrivial cw solutions with identical amplitude but π phase difference.”

From the simulations in FigS18, I do not see any “dark solitons”, but nonlinear fronts which are not locked together to form dark pulses, least with a π -phase shift between them.

The author should thus not refer in the manuscript to the localized structure they observe as “dark solitons”, since the locking mechanism is not between two fronts but instead a pinning effect on the edges of the pulse pump, as explained in the manuscript and well-known in standard Kerr cavity under synchronous pulse driving with opposite sign for the Kerr nonlinearity and the dispersion coefficient.

Response: To avoid confusion, we have replaced all occurrences of “dark soliton” or “dark DQS” with “dark pulse” throughout the revised manuscript, as suggested by the reviewer.

On the other hand, we want to emphasize that in our previous publication (M. Nie, S.-W. Huang PRAp 13 044046 (2020)), the dark DQSs are generated in a degenerate optical parametric oscillator. With parametric gain (including A^* in the equation), the dark DQS should be formed in pairs to accomplish the self-reproduction in the cavity. As for the externally driven case (including A_{in} in the equation), dark DQSs are not required to be formed in pairs.

7/ The discussion of the difference between the RF single-sideband phase noise (SSBPN) of the bright QCSs and fronts, and the comparison with the pump SSBPN, is rather vague. The results suggest that bright QCSs are more prone to noise than fronts and increase the pump noise. However, we would expect the reverse, since cavity solitons, as dissipative structures, should filter out the noise of the pump—at least in some frequency bands—while Fig. S11 shows a much larger SSBPN, which is unusual.

Response: In our setup, the pump repetition rate was not actively stabilized to the cavity FSR. Consequently, relative to the pump pulse, the newly generated comb lines from both the bright DQS and dark pulse exhibit excess phase noise arising from cavity length fluctuations. Furthermore, we observed that the bright DQS exhibits greater sensitivity to pump desynchronization than the dark pulse. Consequently, its phase noise is higher, particularly at offset frequencies below 70 kHz. At higher offset frequencies, however, the phase noise of the bright DQS becomes lower than that of the dark pulse. These contrasting behaviors originate from their distinct pulse formation mechanisms: the bright DQS arises from a double balance between dispersion and nonlinearity, as well as parametric gain and cavity loss, whereas the dark pulse forms through the interlocking of stationary switching waves.

Revisions made: We have expanded the discussion of the SSB phase noise in the revised Supplementary Information Section VI.

VI. Single sideband phase noise spectra of DQS in different states

The single-sideband (SSB) phase noise spectra of DQS in different states are plotted in Fig. S12. Due to the mutual trapping between FF and SH DQS, a good overlap between their SSB phase noise spectra was observed. In our setup, the pump repetition rate was not actively stabilized to the cavity FSR. Consequently, relative to the pump pulse, the newly generated comb lines from both the bright DQS and dark pulse exhibit excess phase noise arising from cavity length fluctuations. Furthermore, we observed that the bright DQS exhibits greater sensitivity to pump desynchronization than the dark pulse. Consequently, its phase noise is higher, particularly at offset frequencies below 70 kHz. At higher offset frequencies, however, the phase noise of the bright DQS becomes lower than that of the dark pulse. These contrasting behaviors originate from their distinct pulse formation mechanisms: the bright DQS arises from a double balance between dispersion and nonlinearity, as well as parametric gain and cavity loss, whereas the dark pulse forms through the interlocking of stationary switching waves.

Fig. S12. SSB phase noise spectra of DQS in different states.

8/ In the perspective of generating two-colour combs via CE-SHG, and beyond the lower power threshold and the control of the sign of the effective third order nonlinearity, are there advantages of generating QCS compared to separating the Kerr soliton formation and the intracavity CE-SHG (bandwidth, efficiency, coherence, etc.)?

Response: To generate dual-color combs with comparable output powers, separating the Kerr-soliton formation and intracavity CE-SHG processes would be considerably more complex and less efficient. (i) The repetition rate of the Kerr soliton must be synchronized with the FSR of the secondary quadratic cavity for CE-SHG. (ii) The Kerr-soliton comb would likely require amplification to effectively drive the CE-SHG process. (iii) The CE-SHG cavity must also be carefully engineered to suppress unwanted cascaded quadratic interactions.

9/ The introduction should be revised, to be more appealing and easier to follow, specifically the structure.

Dissipative solitons were first observed in fiber cavities, this should be properly acknowledged, all the more given the “historical” perspective of the introduction. The recent work of Tang (Opt Lett 49, 2449 (2024)) closely related with the present work should also be cited.

Response: We have added the two references as Ref. [1] (Nat. Photon. 4 (2010), 471) and Ref. [29] (Opt. Lett. 49 (2024), 2449). Following the reviewer's suggestion, we have rewritten the Introduction to make it more concise and engaging.

Revisions made: We have rewritten the first few paragraphs of the Introduction to make it more concise and engaging.

Introduction

A dissipative Kerr soliton (DKS) is a stable, localized wave packet that forms in a dispersion-engineered nonlinear resonator through a double balance between dispersion and Kerr nonlinearity, as well as parametric gain and cavity loss [1,2]. DKS has drawn significant attention for its ability to generate coherent optical frequency combs, enabling transformative advancements across a wide range of fields. Notable applications with DKS microcombs include highly multiplexed coherent optical communications [3,4], astrocombs for precise astronomical spectrograph calibration [5,6], coherent ranging for high-accuracy distance measurement [7,8], dual-comb spectroscopy for fast and precise chemical analysis [9,10], integrated frequency synthesizers [11,12], and optical clock systems [13–15]. Furthermore, their compact size, low power consumption, and potential for on-chip integration make them ideal for portable and space-based technologies.

It has been demonstrated that cascaded quadratic processes can create an effective Kerr nonlinearity (EKN) [16], whose feature can be flexibly controlled via phase mismatch. When both phase mismatch and group velocity mismatch (GVM) are close to zero, the EKN dominates over the material Kerr nonlinearity (MKN). In this regime, EKN becomes the primary nonlinearity governing dissipative soliton formation dynamics, giving rise to the concept of dissipative quadratic solitons (DQSs) [17–24]. Notably, the maximum EKN occurs near the zero phase mismatch point, and its sign can be easily reversed by tuning across this point. This enables a wide range of EKN tuning, from negative to positive, allowing it to balance any dispersion for soliton generation.

In a different regime characterized by large GVM, walk-off-induced DQS has been observed in a synchronously pumped optical parametric oscillator [25]. This form of DQS arises from the interplay among dispersion, gain saturation, timing mismatch, and gain clamping, resulting in a sensitivity to the pump pulse duration. This behavior contrasts with the canonical DQS in the cascaded nonlinearity limit, where the EKN predominantly governs soliton formation dynamics.

Compared to dispersion-engineered DKS, nonlinearity-engineered DQS offers greater flexibility in selecting the comb spectral range and provides *in situ* control over the DQS characteristics. Additionally, the large EKN—typically 100 times greater than the MKN—significantly reduces the threshold power for DQS generation, making it much lower than that required for DKS formation. Finally, since EKN originates from the two-wave interaction in the quadratically nonlinear crystal, DQS is intrinsically a pair of dual-color mode-locked pulses and can be easily extended to form multi-color combs via the efficient intracavity nonlinear frequency conversion.

Recent experiments have offered encouraging insights into the development of DQS [26–30]. Using periodically poled lithium niobate (PPLN) crystals, modulation instability (MI) frequency combs have been observed [26–29], and the demonstration of quantum-correlated twin beams has marked a significant milestone [30]. Advancing from MI combs to DQS generation, which promises higher coherence and broader bandwidth, remains a challenging frontier.

In this paper, we demonstrate nonlinearity-engineered DQS in a free-space, singly resonant cavity-enhanced second-harmonic generation (CE-SHG) setup (Fig. 1). A single mean-field equation is derived to describe the DQS formation dynamics, highlighting the origin of the large effective EKN. The DQS nature is identified through excellently matched simulation and experiment, revealing a dual-color pulse pair with sech²-shaped spectrum for the fundamental field (FF) and a flat-top spectrum for the second harmonic (SH) field. Initially, the phase mismatch is set to a negative value, allowing the highly negative EKN to balance the normal dispersion of the PPLN crystal at 1571 nm, enabling ultralow-threshold bright DQS generation in the normal dispersion regime. By simply adjusting the crystal temperature, we reverse the phase mismatch and consequently the EKN, achieving *in situ* switching between bright DQSs and dark pulses, each with distinct spectro-temporal characteristics [31–34]. Additionally, by tuning the pump wavelength and crystal temperature, we demonstrate *in situ* control over DQS comb properties. Finally, besides SH combs centered at 786 nm, other visible combs centered at 524 nm are also generated through intrinsic sum

frequency generation (SFG), offering potential applications in comb self-referencing, optical atomic clocks, and quantum information science [35–39].

10/ The quality of the figures is poor and looks like assembled screenshots. All figures should be labelled. The axis label should be consistent e.g. not mix “Bandwidth” and “Wavelength”. All curves should be discussed in the caption. CTWE, SMFE should be defined or referred to equations in the main manuscript or SI.

Response: We believe the figures are compressed with poor quality in the Manuscript Submission System and the issue should be solved in the final editorial step. Following the reviewer’s suggestion, we have labelled all the figures and unified the axis labels of Figs. 3 and 4 in the revised manuscript. SMFE and CTWE were previously defined only in the text; following the reviewer’s suggestion, we have added their definitions to the caption of the revised Fig. 2.

11/ Other comments:

- A lot of acronyms are used thorough the paper, making it quite difficult to follow.

Response: In the revised manuscript, we have reduced the use of acronyms and added their corresponding definitions, particularly in the figure captions throughout the manuscript.

- There is a confusion about (ξ) and the wave-vector mismatch parameter. In the manuscript, it is said: “ $\xi=\Delta kL$ is the wave-vector mismatch parameter”; or “When the wave-vector mismatch $\xi=m\cdot 2\pi$ ”; but also “ Δk is the wave-vector mismatch”. ξ is not a wave vector mismatch. Please revise for correctness and consistency.

Response: Following the reviewer’s suggestion, we now refer to ξ as “phase mismatch” and Δk as the “wave-vector mismatch” throughout the manuscript for clarity and consistency.

- Please do not use the term “soliton pair” when referring to the FF-SH locked dissipative structures, given the conventional meaning of a soliton pair in the context of dissipative solitons. Moreover, the SH pulse does not exhibit solitonic behavior and should not be referred to as a soliton.

Response: Following the reviewer’s suggestion, we have replaced “soliton pair” with “pulse pair” throughout the manuscript.

- In Fig1. The word “Length” is not appropriate. In the bottom right inset, I suggest replacing “Large and negative NL” by “effective negative Kerr nonlinearity”, which is more specific to this work.

Response: Following the reviewer’s suggestions, we have replaced “Length” and “Large and negative NL” with “Propagation distance” and “Large and negative effective Kerr nonlinearity,” respectively.

-The first sentence of the Discussion section should be revised.

Response: We have removed the phrase “the first” from the opening sentence of the Discussion section. The revised sentence now reads: “In conclusion, we have provided a deeper theoretical understanding of DQS dynamics through nonlinearity engineering and successfully demonstrated DQS in a free-space, singly resonant, cavity-enhanced second-harmonic generation setup.”

Reviewer #3 (Comments to the Author):

The authors studied dissipative quadratic cavity solitons in free space and with the infrared pump beam resonance only. This experimental study also relies on a unique mean-field equation, derived to describe dissipative quadratic cavity solitons and extract the dynamics of the resonant beam, thus producing a pair of two-color pulses. Phase-mismatching conditions allow switching between bright and dark solitary waves, each exhibiting distinct spectrotemporal characteristics. Thus, in addition to SH combs centered near the second harmonic position, other visible combs centered at lower wavelengths are also generated by intrinsic sum frequency generation.

The work shown in that paper is of significance in the field of quadratic soliton cavity and the results are good and convincing.

The experimental demonstration and theoretical approach are well presented and are in good agreement. The methodology is good and the work is sufficiently well described.

The article is of good quality.

Remark : The first theoretical published paper on quadratic solitons is not the one referenced in [17] but the one published by Karamzin and Sukhorukov in 1974 (Theoretical prediction). The authors could add this reference “Karamzin, Y. N., & Sukhorukov, A. P. (1974). Nonlinear interaction of diffracted light beams in a medium with quadratic nonlinearity: mutual focusing of beams and limitation on the efficiency of optical frequency converters. *ZhETF Pisma Redaktsiiu*, 20, 734.”

Response: Thanks for pointing out the correct reference. On the other hand, we decide to remove all the references and claims related to spatial solitons to avoid confusion and improve clarity. We have also rewritten the Introduction to make it more concise and engaging.

Q1: Could the authors explain why they used a PPLN in a type 1 phase matching and not a type 0 phase matching?

Response: As laid out in the **Operating principles**, near-zero group velocity mismatch (GVM) is the key to obtaining dissipative quadratic cavity soliton. We employed type-I phase matching to achieve the near-zero GVM condition, as described in the **Bright DQS generation** section: “To achieve near-zero GVM across the bandwidth of erbium-doped fiber amplifier (EDFA), we employed type-I phase matching configuration ($o+o\rightarrow e$, nonlinear coefficient of $d_{eff}=2.7$ pm/V) in the bulk PPLN crystal, where the zero-GVM wavelength is at ~ 1571 nm (see Fig. S7 in Supplementary Information Section III)”. Fig. S7 plots the GVM as a function of FF wavelength at different type-I PPLN crystal temperatures, showing the near-zero GVM condition can be satisfied across the C and L bands. In contrast, the figure on the right shows that the GVM exceeds 250 fs/mm when type 0 phase matching is used, which is too large for supporting DQS formation.

Q2: Could the authors explain why they don't try to make the second harmonic resonant?

Response: Reviewer 3's intuition is correct. According to our theory, a doubly resonant system can indeed operate with a lower threshold, making continuous-wave pumping feasible. However, we were unable to identify a coating provider capable of ensuring high reflectivity at both 1570 nm and 785 nm.

Q3: The shape of solitons in the time domain is not presented in the main article. It would be interesting to present both the spectral and temporal evolutions in the same figure.

Response: Following the reviewer's suggestion, we have added a new Supplementary Information Section VII that presents the intensity autocorrelation traces of both the bright DQS and dark pulse. In addition, we have included the simulated temporal traces in Figs. 3 and 4 of the main text, in response to the other reviewers' comments.

Revisions made:

VII. Intensity autocorrelation of bright DQS and dark pulse

Figure S13a shows the optical spectrum of the single bright DQS after removing the pump background with a volume Bragg grating (VBG), and Fig. S13b presents the corresponding intensity autocorrelation trace, confirming its femtosecond pulse characteristics. Figure S13c shows the optical spectrum of the dark pulse in state I at a normalized detuning of 5.7, and Fig. S13d presents the corresponding intensity autocorrelation trace, confirming its flat-top temporal profile.

Fig. S13. (a) Optical spectrum of the single bright DQS with the pump background removed by VBG. (b) The corresponding intensity autocorrelation trace. (c) Optical spectrum of the dark pulse in state I with a normalized detuning of 5.7. (d) The corresponding intensity autocorrelation trace.

Fig. 3. Experimental generation of bright DQS. (a) Output average power of the FF and SH as a function of normalized detuning $\Delta_1 = \delta_1/\alpha_1$. (b) Optical spectra of the bright DQS at the FF with different normalized detunings. (c) Optical spectra of the bright DQS at the SH with different normalized detunings. (d) RF spectra of the bright DQS at the FF with different detunings. (e) Optical spectrum (left) and RF spectrum (right) of the single bright DQS at the FF, taken with a normalized detuning of -7.6. (f) Optical spectrum (left) and RF spectrum (right) of the single bright DQS at the SH, taken with a normalized detuning of -7.6. The spectral shoulder beyond 1580 nm originates from the amplified spontaneous emission of the L-band erbium-doped fiber amplifier. The identical repetition rates of the FF and SH confirm the mutual trapping of the dual-color bright DQS. (g)-(i) The effect of GVM on the optical spectrum at the FF and SH. The third column shows the corresponding simulated temporal profiles. (g) GVM=0 fs/mm, $\Delta_1 = -7.6$; (h) GVM=5.5 fs/mm, $\Delta_1 = -7.0$; (i) GVM=11 fs/mm, $\Delta_1 = -6.8$. (j)-(l) The effect of phase mismatch on the optical spectrum at the FF and SH. The third column shows the corresponding simulated temporal profiles. (j) $\xi = -1.4\pi$, $\Delta_1 = -6.4$; (k) $\xi = -2.0\pi$, $\Delta_1 = -7.6$; (l) $\xi = -2.4\pi$, $\Delta_1 = -7.6$.

Fig. 4. Experimental generation of dark pulse. (a) Output average power of the FF and SH as a function of normalized detuning $\Delta_1 = \delta_1/\alpha_1$. (b) Optical spectra of the dark pulse at the FF with different normalized detunings. (c) Optical spectra of the dark pulse at the SH with different normalized detunings. (d) RF spectra of the dark pulse at the FF (top) and SH (bottom) in state I with a normalized detuning of 5.7. The identical repetition rates of the FF and SH confirm the mutual trapping of the dual-color dark pulse. (e)-(g) The effect of GVM on the optical spectrum at the FF and SH. The third column shows the corresponding simulated temporal profiles. (e) GVM=0 fs/mm, $\Delta_1=7.0$; (f) GVM=5.5 fs/mm, $\Delta_1=6.8$; (g) GVM=11 fs/mm, $\Delta_1=6.2$. (h)-(j) The effect of phase mismatch on the optical spectrum at the FF and SH. The third column shows the corresponding simulated temporal profiles. (h) $\xi=1.5\pi$, $\Delta_1=6.6$; (i) $\xi=2.0\pi$, $\Delta_1=7.0$; (j) $\xi=2.4\pi$, $\Delta_1=5.5$.

Q4: Nonlinear conversion affects the time domain but can also introduce spatial distortion. Did the authors observe any trapping or spatial broadening of the beam?

Response: We appreciate the reviewer's insightful question. We did not observe any trapping or spatial broadening of the beam. In our experiment, there is continuous-wave component due to low extinction ratio during the pulse picking process. Therefore, the average power of the soliton pedestal is much higher than that of the soliton pulse itself. Since only the high-peak power soliton pulse can induce beam trapping or broadening and the common camera is a low-bandwidth device (~ 1 kHz) that can only response to the average power, the optical intensity felt by the camera is dominant by the continuous-wave component instead of the soliton pulse. To observe the beam trapping or broadening, a much faster camera and a much better pulse picking setup are required. This is something worth studying as a follow-up study.

Reviewer #2 (Remarks to the Author):

The authors have satisfactorily addressed most of the reviewer's concerns. However, though the results are of interest and deserved to be published, I still have several comments that prevent me from supporting the publication of the manuscript in its current form.

1/ In their revised version, the authors continue to employ terminology that is inappropriate for describing the dissipative structure observed in the normal dispersion regime (see point 7 of Rev2 comments). True dark pulses with synchronous pumping in normal dispersion Kerr cavities have been reported with a clear dip in the pulse center (see Garbin, Experimental and numerical investigations of switching wave dynamics in a normally dispersive fibre ring resonator 2017).

On the contrary, Fig4 of the manuscript shows the formation of BRIGHT pulses, which cannot thus be called "dark pulses". The authors must change how they called this dissipative structure.

Response: Following the reviewer's comment, we have changed "dark pulse" to "platicon" throughout the manuscript to be consistent with the terminology used in the community [Opt. Lett. 38, 3899 (2013), Opt. Express 23, 7713 (2015), Nat. Commun. 13, 1771 (2022), and Photon. Res. 10, 1877 (2022)].

2/ In the revised introduction (and in Supplemental Information section VI), the first sentence is not correct. The loss experienced by the soliton is balanced by the coherent driving, not by "parametric gain" in dispersive Kerr resonators. Note that parametrically driven Kerr cavity solitons, which are sustained by the equilibrium between parametric (phase sensitive) gain and loss, also exist and represent distinct types of solitons.

Response: The first sentence of the Introduction refers to DKS, and therefore the description is correct. On the other hand, the reviewer is correct that the balance between "coherent driving" and cavity loss provides a more accurate description of the DQS in CE-SHG. Following the reviewer's suggestion, we have replaced "parametric gain" with "coherent driving" in the caption of Figure 1 where it appears.

3/ The authors explain in the revised caption of Figure3 that the asymmetry is due to the amplified spontaneous emission in EDFA. The existence of an EDFA in the experiment must be mentioned in the Method section and in the Supp Info Section about the pumped laser (not just mentioned in the caption of a figure).

Response: Following the reviewer's suggestion, we have mentioned the EDFA in the Method section and provided the amplified pump spectrum in Figure S5.

4/ Fig5c: the labels of the axes are not appropriate: horizontal is "delay" and vertical is "Autocorrelation signal".

Response: We have changed the x-axis label to "Delay (ps)" and y-axis label to "Autocorrelation Signal".

5/ The answer about the SSB phase noise does not convince me. Experiments are reported in the literature about (long) synchronously pumped cavities without repetition rate active stabilization.

It is expected the soliton, as a dissipative structure in a cavity to filters the jitter of the noise and thus the comb exhibits lower SSB phase noise. One explanation could be a jitter induced by the noise injected by the ASE of the amplifier in the pump laser source.

Response: We appreciate the reviewer's insight. While we agree that ASE can be a possible explanation of the excessive phase noise, we are not sure if it is the dominant factor since the linear filtering effect of the cavity will reject most of the ASE noise. Without a dedicated and comprehensive follow-up study, we are not in a position to offer a conclusive explanation of the phase noise dynamics. Therefore, we have revised the discussion in Supplementary Information Section VI to focus on the factual results from the phase noise measurements.

Furthermore, the phase noise of both the bright DQS and the platicon is higher than that of the pump. One possible explanation is that the excessive phase noise arises from cavity-length fluctuations, as the pump repetition rate was not actively stabilized to the cavity FSR in our setup. Another possible explanation is that the excess phase noise is induced by the injected ASE of the L-band EDFA (Fig. S5c). A dedicated and comprehensive follow-up study is required to elucidate the physical origin of the observed phase noise behavior.